# Automatic large-scale political bias detection of news outlets

**Ronja Rönnback**[ID]*, **Chris Emmery**[ID], **Henry Brighton**

Department of Cognitive Science and Artificial Intelligence, Tilburg University, Tilburg, The Netherlands

* r.g.i.ronnback@tilburguniversity.edu

## Abstract

Political bias is an inescapable characteristic in news and media reporting, and understanding what political biases people are exposed to when interacting with online news is of crucial import. However, quantifying political bias is problematic. To systematically study the political biases of online news, much of previous research has used human-labelled databases. Yet, these databases tend to be costly, and cover only a few thousand instances at most. Additionally, despite the wide recognition that bias can be expressed in a multitude of ways, many have only examined narrow expressions of bias. For example, most have focused on biased wording in news articles, but ignore bias expressed when an outlet avoids reporting on certain topics or events. In this article, we introduce a data-driven approach that uses machine learning techniques to analyse multiple forms of bias, and that can estimate the political leaning of hundreds of thousands of Web domains with high accuracy. Crucially, this approach also allows us to provide detailed explanations for why a news outlet is assigned a particular political bias. Our work thereby presents a scalable and comprehensive approach to studying political bias in news on a larger scale than ever before.

## Introduction

The proper functioning of a democratic system presumes that its citizens have the tools to make well-informed decisions. Yet, bias in news is unavoidable. Understanding how people are exposed to political bias when interacting with Web technologies like social media, search engines, or other sources is crucial for contributing to better-informed societies. This can be especially important when the bias of the source cannot easily be anticipated, such as when the source is unfamiliar. It raises the concrete question of how to study news bias. An immediate challenge lies in how to measure political bias in the first place; most would agree that Fox News and the Guardian behave differently and occupy different locations on the political spectrum. However, the assumption of a left-to-right political spectrum is by no means uncontroversial [1–3]. It is a simplifying one that many (our work included) choose to make to be able to systematically study political biases in online services. Many researchers measure bias by using such labels, which detail properties like reliability or political leaning of news. These labels are generally derived in one of two ways: human labelling or computational labelling.

**Data availability statement:** We use openly available datasets available via https://www.gdeltproject.org/data.htmlfortheGDELTdataset, https://mediabiasfactcheck.com/mbfcs-data-api/

for the Media Bias fact Check data, and https://personalization.ccs.neu.edu/Projects/Partisanship/ for Robertson et al.'s data. Our code used to process these datasets is available at https://github.com/rtronnback/automatic_news_monitoring_with_GDELT.

**Funding:** The author(s) received no specific funding for this work.

**Competing interests:** The authors have declared that no competing interests exist.

Human labelling can allow researchers to deal with ambiguity and contextual information, but this approach is often slow, laborious and, ironically, can be subject to bias as well [4]. On the other hand, data-driven approaches enable fast and efficient analysis of news through Natural Language Processing (NLP) and Machine Learning (ML), yet fail to provide the same level of insight as manual labelling, falling short of actually increasing understanding of the phenomenon [4].

Moreover, many simplify even further by focusing on narrow expressions of bias in, for instance, word choice in headlines. Yet, some forms of bias cannot be detected unless examining the holistic behaviour of a news outlet, rather than individual news items. For example, an outlet systematically avoiding a topic, or only covering it very briefly despite societal relevance, can be a clear sign of bias. Yet, this is often not considered in existing research.

Given these two challenges, our work proposes an approach that uses automatic labelling of news web-domains' bias on a global scale. To do this, we use ML to predict web-domain bias using the Global Database of Events, Language, and Tone (GDELT). GDELT tracks and analyses global news, making it an ideal source for this task [5]. It enables us to: i) focus on aggregated outlet information rather than article-, sentence- or word-level analysis (predominant in related work), ii) differentiate between multiple types of bias to review their impact, whereas much of previous research studies one sub-type, and iii) evaluate model performance against the provenance of the true bias labels (either computationally derived or human-annotated). Finally, we combine these computational methods with techniques for model explainability to extract the approximate reasoning behind why a news web-domain in question is deemed to be politically biased. We believe our analyses may prove meaningful for establishing recurrent problematic behaviour on the part of news outlets in an automatic manner, and, if developed further, could help citizens inform themselves as to the partiality of their news sources.

## Barriers of news bias studies: Narrow focus, scope and lack of insight

Political bias is challenging to define and more often than not considered to be subjective, but generally refers to a recurring (intentional or unintentional) attempt to influence a reader [4, 6]. There are many ways that bias can manifest itself in news media. This can range from the selection of what events to cover, where an article should be placed on the homepage, how much space to give it, or whether to (as the classic example goes) refer to "freedom fighters" as opposed to "terrorists" [4,7]. As a consequence, measuring bias raises a lot of practical problems, and many have focused on studying very specific types of bias to simplify the task. What follows is a succinct overview of biases identified in Hamborg, Donnay and Gipp's [4] literature review, and that we focus on in our analyses.

- **Event selection bias** or coverage bias involves choosing which events merit report. Naturally, not all stories can, nor should be published. Yet, intentional and consistent avoidance of or focus on a topic can influence or mislead audiences. This is a well-studied phenomenon in crime reporting [8–10].
- **Labelling** and **word choice bias** are a major focus of study. This concerns framing events or highlighting a certain perspective by choosing labels or particular words that, while similar, will convey different meanings to audiences: for example, referring to something as a "special military operation" or "intervention" instead of "invasion" may change perceptions of events.

- **Size allocation bias** concerns the length of articles. The amount of text written on some topics may introduce certain outlet biases. For example, it is possible that news outlets report consistently but only at brief length on certain topics while dedicating a lot of work and space to others. This is a relatively straightforward form of bias to study, though it has not received much attention [4].
- **Picture selection** and **explanation bias** concern what pictures are chosen to accompany certain articles, and how those pictures are described. Images have been shown to affect readers' perceptions of news articles [11,12], therefore selecting and describing them is susceptible to potential biases.

These subtypes of bias have, to varying extents, been examined in previous literature. However, it is rare that a single study encompasses more than one form of bias. Furthermore, many have focused on making article-level inferences (trained directly on the *content* of the articles, and therefore often fixating on word choice bias), rather than outlet-level inferences (based on *meta-data* of multiple articles, which could encompass multiple of the bias subtypes outlined above).

On article-level, studies have used computational (NLP) tools such as Term Frequency-Inverse Document Frequency [13,14] or doc2vec [15] as feature representation methods, but many have had limited success [16,17] and rely on costly resources [16–18]. Gangula et al. [18], for example, aimed to predict news bias towards five local political parties based on headlines, articles and a combination of the two. They achieved an accuracy of 89% with an attention-based model. However, the narrow focus and reliance on very specific human annotations limits the work's ability to scale to a wider context and to provide deeper understanding of political bias on the whole. Spinde et al. [19], on the other hand, use existing labelled datasets from Reddit comments, movie reviews, Wikipedia, and two general language datasets. These were combined to train a DistilBERT model [20] in a Multitask Learning setting. While the results look promising (F1-score of 0.77), their results are only partially transferable to news due to the data being only indirectly related to news bias, as they themselves note. A follow-up study compiled a dataset of 3,700 sentence-level expert annotations on a broad range of topics in lieu of the usual crowd-sourced annotations. BERT-based models [21] detected sentence bias, achieving a maximum F1-score of 0.80 [22]. This constitutes an improvement, though backs off to extensive manual annotation and still limits the focus to word choice bias on an article-level. Finally, some previous work has focused on detecting a dramatically wider range of broad bias subtypes on sentence-level (ad hominem or circular reasoning bias, for example) [23]. These distinctly focus on political bias as a subtype, however, rather than as a nuanced subject that can be expressed in a number of different ways [4], as the current work does.

Not all existing work focuses on article- or sentence-level bias, or even uses ML to estimate website or news political bias, however. For example, using the Twitter accounts of users who were registered as either Republican or Democrat voters, Le, Shafiq and Srinivasan [24] approximated bias based on how often users shared articles from outlets or websites. Articles shared frequently by Republicans would thus be assumed to stem from web-domains with a right-wing political leaning, and vice versa. Given this method, Robertson et al. [25] assigned and validated political leaning scores for over twenty thousand websites. This approach scales well and provides follow-up studies with validated political bias scores. Nevertheless, it presents an approximated measure of bias and does not delve deeper into what makes a particular outlet more biased towards a political audience.

Work that bears resemblance to our own is MediaRank [26], which also opts for a source-level analysis to create quality rankings of the world's most prominent news sources. Using

metrics such as reputation, reporting bias, financial pressure, and popularity, they evaluate over 50 thousand news sources in 68 countries. There are some important distinctions, however: firstly, they rely on an outlet's average sentiment regarding celebrity Democrats and Republicans. In contrast, our approach makes use of the millions of GDELT themes and ready-made NLP features that have been selected in an entirely data-driven fashion. This makes our approach more comprehensive and globally applicable (also for alternative divides of the political spectrum), since American politicians or celebrities will not be extensively discussed everywhere. As such, our project can provide an assessment of news sources that is more focused on an in-depth examination of political bias using multiple related features and themes. Therefore, while MediaRank focuses on a generalist perspective of news quality, our approach offers a more in-depth study of political bias, the numerous ways it can manifest, and does this at a globally applicable scale.

It is worth noting that large language models (LLMs) present a promising avenue for large-scale sentence- and article-level bias classification. LLMs offer a low barrier of entry to interfacing with large amounts of textual data, and especially to extract information from complex structures. They have therefore been used to for example automatically detect misinformation or fake news [27–29], and various forms of bias [28,30,31], and might therefore be deemed as relevant for our project as well. However, LLMs-based approaches face some serious challenges. Namely, LLMs are subject to various internal political biases [32,33], and seem to consistently differ from human judgement [27], which presents a serious complication for LLM-based applications [34]. Additionally, their inherent stochasticity implies that the accuracy of outputs may differ significantly despite receiving the same prompts, and has been shown to produce contradicting results when dealing with political disinformation [29]. Previous work has also aptly noted that LLMs do not receive regular updates, and that this may present a problem in the rapidly evolving news cycle [31]. Finally, the quality of LLM classifications has also been shown to not match the performance of fine-tuned supervised models on numerous applications (social understanding [35], media bias detection [28], as well as other social science tasks [36,37]). Despite these limitations, we provide two naive zero-shot LLM baselines for comparative purposes. This is relevant given the aforementioned potential of LLMs to replace manual labelling and ease of use, but importantly also allows us to examine if the performance limitations found in previous work are repeated here. It is worth noting that, while LLMs offer advantages like ease of use, there are also trade-offs to consider, such as, hardware costs, output fabrication, and energy consumption, especially if used on a large scale, among other challenges [38]. Appendix E details the implementation and prompting used.

The aforementioned examples all demonstrate the potential of data-driven news bias detection but also the existing limitations. Focus on article- or sentence-level bias excludes important patterns that emerge from a news web-domain's behaviour as a whole, such as coverage, story placement and size patterns (i.e., are some topics avoided, only given limited space, or only reported upon very briefly). These could play a key role in demonstrating and explaining bias. Additionally, many studies rely on costly expert or crowd-sourced annotations [18,22], or on tangential datasets [19].

Crucially, to identify media bias of news outlets at scale, it is not sufficient to rely on specific topics, small-scale datasets or only on sentence- or article-level classifications. Instead, news bias monitoring should ideally cover a multitude of topics and be applicable to (nearly) any web-domain, whether this be from well-known sources such as breitbart.com or cnn.com or from lesser known organisations. Therefore, our work aims to develop a system that relies on a broader data source which specifically covers news, facilitates global-scale coverage, and is capable of examining multiple facets of media bias more thoroughly. As a result, this

approach is entirely data-driven and reliant on automated techniques, rather than time-consuming manual approaches. We believe this could hugely benefit the field and ensure a holistic coverage of news bias.

## An approach for large-scale bias labelling

To make inferences about web-domains on a global scale, one of course needs a data source with global coverage. This is possible thanks to the Global Database of Events, Language, and Tone (GDELT), as well as some features from the independent media bias tracking organisation Media Bias Fact Check (MBFC). For further details of implementation, please refer to Appendix B.

**GDELT** [5] is an open platform for monitoring global news, and the most extensive database covering news in existence, to the best of the authors' knowledge. It has been used in related previous work examining the use of images in news [39], as well as the rise fake news [40]. Compared to similar datasets, it has been found to contain a broader set of unique news outlets [41], making it ideal for our application. The main dataset of interest here is the Global Knowledge Graph (GKG), which records "latent dimensions, geography and network structure of the global news" [5, p. 1]. It contains various automatically identified themes associated with articles (these are extensive, covering topics ranging from immigration to gasoline prices or even specific currencies or mammals) and the results of various analyses, such as tone. Given the size of the dataset (one year's worth of data corresponds to 2.5TB [42]), our experiments use a limited sample consisting of English articles from the year 2022. This excludes articles that have been translated, and subsequently focuses on Western countries which approximately follow the bipartisan political spectrum. Though there may be some slight selection bias given the outbreak of the COVID-19 pandemic, we expect that the breadth of GDELT alleviates concerns of generalisability. We filtered all features that we deemed unlikely to reflect bias, resulting in features detailed in Table 1.

We found more than 30,000 unique themes in our sample. To reduce their sparsity, we opted to filter over-specific taxonomic items (e.g., specific birds, mammals, or fish), and other themes appearing less than 1000 times and more than one standard deviation from the mean of the log transform of theme frequency. In addition to maintaining a data-driven approach, this method does not limit the sample to known politically controversial themes

**Table 1. GDELT features included in analysis, accompanied by short description.**

| GDELT Feature | Description |
| --- | --- |
| Tone | The positive and negative tone of the article as a whole. |
| Polarity | Proportion of words that matched a tonal dictionary to indicate how polarized the text is. For example, a high polarity but similar scores for positive and negative tones indicates that the article contains roughly the same amount of positively and negatively charged words. |
| Activity reference density | Percentage score of active words, like active verbs, in the article and is supposed to act as a proxy for "activeness" in the text as compared to merely descriptive text. |
| Self or group reference density | Percentage of pronouns present. The GDELT documentation states that this "can be used to distinguish certain classes of news media and certain contexts" [43]. |
| Word count | Number of words in the article. |
| Visual content | Presence of cover images, embedded social media images or videos related to the article. This was noted merely as either "present" or "absent" for the purposes of out work, |
| Themes | A list of the themes detected in the article as by GDELT's analysis. |

(such as immigration, abortion, or climate change [22,44]) thereby potentially including under-explored indicators of bias.

Additionally, to account for other forms of bias beyond those found in tone or sentiment analyses, we added a set of outlet-level features per theme: the proportion of articles (as this might allow for examination of selection bias, and whether some topics are ignored or excessively focused on), the average word count per article (which might reveal size bias), and the presence of images or videos (which could also be indicative of under-explored forms of bias). Crucially, this aggregation based on themes makes it easy to notice if some theme is largely ignored or alternatively excessively focused on by a source (for example, if a source very rarely and only briefly covers news related to global warming, the proportion and average word count of articles for that theme would be very low). This preprocessing yields one row per web-domain, with associated bias features from GDELT per theme. Due to the high number of features this resulted in, we also eliminated some features (for details, see Appendix B). Fig 1 provides an overview of these preprocessing steps.

Aside from GDELT, we also used some information from Media Bias Fact Check (**MBFC**). MBFC provides information about the factuality, traffic, country of origin, press freedom, media type, and credibility of news web-domains. As such information is relevant but not available through GDELT, we extended the training dataset with these features (see Appendix A for a full explanation of the features). These additional features were however only included in one of the experiments, as is further detailed in the sections below.

## Labelling: Human reliance or fully automated?

Given the laborious nature of human labelling, it would be cheaper and faster to rely on automatic methods, but only if performance is somewhat comparable. To better examine this, we considered model performance on two datasets of ground truth labels. One is composed of human evaluations from MBFC, whereas the other is automatically derived by Robertson et al. [25].

MBFC is an independent organisation that estimates media bias based on human evaluations [45]. Their main aim is to promote awareness of bias and misinformation. For each news web-domain, they provide a political leaning label and some other noteworthy metrics (these are described in detail in Appendix A). The bias label is determined based on a set of topics like, to only name a few, immigration, economic policy, or social views [44]. They do however note that these topics are derived from an American perspective, and may therefore

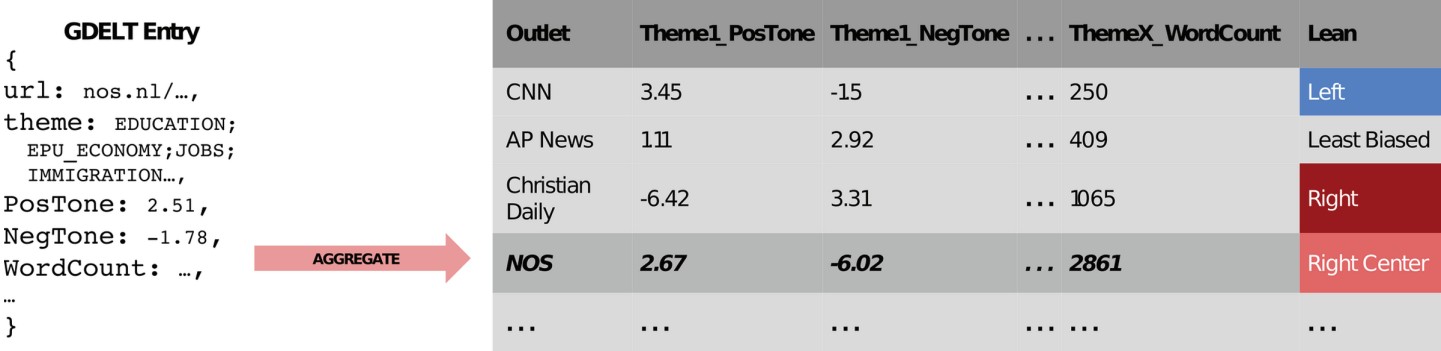

**Fig 1. Overview of data aggregation process.** Overview of process to aggregate GDELT data from article- to outlet-level instances, containing themes and their respective average GDELT features.

not perfectly apply to all countries. While there are numerous entities collecting similar ratings of political media bias and we acknowledge that no ground truth is perfectly unbiased or accurate, MBFC's methodology is thoroughly documented [46], and their dataset is open and extensive (Fig 2). Their labels serve as a ground truth value, and has been used for this same purpose in previous work as well [26].

The other set of labels stems from research by Robertson et al. [25], who built a dataset of bias scores for nearly twenty thousand websites (henceforth referred to as PABS, retrieved from github.com/gitronald/domains/tree/master/data/bias_scores). By relying on Twitter users who were officially registered as either Republican or Democrat voters, they collected all links to web-domains that these users shared on the platform. Operating on the assumption that users would predominantly share links to domains they agreed with, they create a proxy score of the political bias of a web-domain based on the proportion of times it was shared by Democrat versus Republican users. Scores range from -1 to 1, wherein -1 indicates that the source was shared exclusively by Democrats (left-learning bias), and a score of 1 indicates it was shared exclusively by Republicans (right-lean bias). Automatically derived scores such as these are cheaper to obtain but might not be as accurate as human-made labels. We examine whether this is the case by comparing whether results differ between such proxy-labels and human-made labels. The data was preprocessed by binning the continuous values into the five bias classes.

## Classification with machine-learning models

We trained various ML models to classify web-domain bias based on the preprocessed GDELT features, using either MBFC or PABS data as a ground truth [25]. Serving as a point of comparison, a majority baseline model was implemented (i.e., one that invariably classifies all instances as the most common class: "least biased"). The other models included a feed-forward neural network that was trained using Pytorch [47], and Support Vector Machine (SVM, [48,49]), AdaBoost [50,51], and XGBoost classifiers [52] We lastly also include two

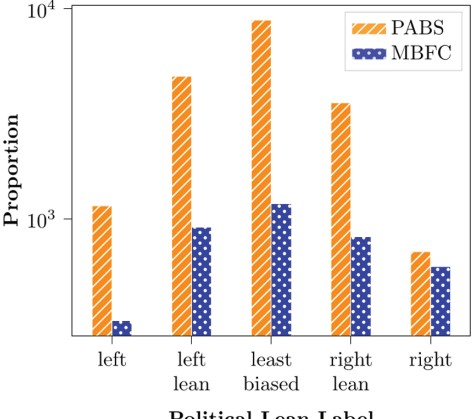

**(a)** Frequency of political lean classes per dataset, on a logarithmic scale.

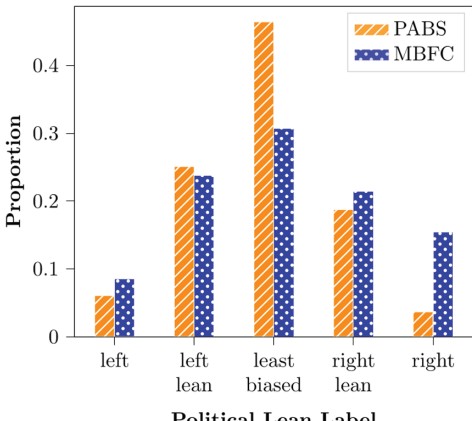

**(b)** Proportion of political lean classes per dataset.

**Fig 2.** Frequency of political lean classes per dataset, on a logarithmic scale.

baselines using a naive zero-shot LLM with `Llama 3.1`[53] and `GPT-4o mini`[54], detailed further in Appendix E.

We empirically determined a two-linear-layered network to perform best (with ReLU activation, batch normalization and a dropout-rate of 0.5 between each layer [55–57]). Training was done via the Adam optimizer [58] and negative log-likelihood loss. Additionally, as one of the experiments involved categorical variables from MBFC, we embedded these [59,60]. The complete structure of the networks can be found in Appendix C. All other models are trained using ten-fold cross-validation and optimized using halving grid search [61] for tuning; the hyper-parameters used in the grid search are detailed in Appendix B and C.

## Model explainability for insight into web-domain bias

Hamborg, Donnay and Gipp [4] criticize computational analyses of media bias of lacking insight into how bias is manifested. Indeed, numerous previous studies have merely focused on determining whether it is present [16,18,19,22]. We therefore opted to address this through the use of computational methods that provide explanations of model decisions. Specifically, we used Shapley Additive Explanations (SHAP), which expands upon six pre-existing methods [62] (examples include LIME [63] and Layer-wise Relevance Propagation [64]). The SHAP framework provides model-agnostic explanations, meaning it can be applied to traditionally inscrutable black-box models. SHAP averages the differences in the model's output with and without a particular feature; the resulting set of differences is then used to approximate the Shapley values for each feature, representing the contribution of that feature to a prediction. This allows us to provide thorough outlet-specific explanations rather than simply model-level insight (as is the case for other traditional explainability frameworks). Consequently, we can scrutinize any news domain to understand why a model classifies it as left- or right-wing biased, providing direct insight into the manifestation of bias and therefore addressing the critique of Hamborg, Donnay and Gipp [4].

## Experimental setup and testing

Our first three experiments were repeated using either MBFC (human-made labels) or PABS scores (automatically derived labels) as ground truth labels. This analysis served to determine whether either labelling method is more successful; as automatically derived labels are easier to obtain but may be less accurate, such a comparison is informative. Finally, a post-hoc analysis examined the difference between the bias labels by MBFC and PABS. The code necessary to replicate our experiments is available at github.com/rtronnback/automatic_news_monitoring_with_GDELT. The following paragraphs provide an overview of each experiment:

**Traditional Bias Experiment** trained models on data related solely to word bias, meaning it covered features related to tone, polarity, activity- and self/group reference density— features that have been more extensively studied in prior work.

**Alternative Bias Experiment** used word-, article-counts, and image or video presence, aiming to better glean the significance of lesser-studied forms of bias; namely, size, selection, and picture bias respectively (the content of the images is not accounted for, thus this feature only approximates picture bias).

**Full Bias Experiment** used all features. This structure aimed to allow for a better examination of the information value of different forms of bias and to extend the analysis beyond the traditional focus of word bias.

**Full Bias & Categorical Features Experiment** was conducted adding various categorical features provided by MBFC such as credibility, factuality ratings, traffic estimates, country

press freedom index, and media type (see Appendix A for full list). We expected these features to be informative and thus conducive to improved performance. This experiment was only done for the full bias dataset using MBFC as the ground truth. For an overview of all experiments, see Fig 3.

## Results

Models were evaluated based on how well they predict a news web-domain's political bias, and results are detailed in Table 2. The overall best-performing model was the neural network trained on the full dataset supplemented with categorical features of MBFC (see Appendix A for the full descriptions). Some examples of news domains, the ground truth and the model's predictions are shown in Table Table 3. It classified web-domains with an accuracy of 76%, and an AUC score of 81%, compared to the baseline model which achieved 45% and an AUC of 50%. The LLM baseline performed similarly to this naive baseline. Models using multiple manifestations of bias generally achieved better performance compared to those using traditional or alternative forms of bias only. The confusion matrices of the best performing model under each experimental condition (traditional bias, alternative bias or both) are shown in Fig 4. Models trained on MBFC as ground truth outperformed models trained on PABS, which achieved only a maximum accuracy of 58.2% and an AUC of 70% with the neural network trained on the alternative bias dataset.

We performed some simple error analysis to examine the strengths and limitations of the best performing model. Detailed results can be found in Appendix F. Considering the different classes, the model performs best at classifying right-wing sources, followed by least biased, and right-centre. Left and left-leaning results were harder for the model to correctly detect. Furthermore, the error rates of the model were lowest for outlets with minimal and medium traffic. This is interesting given that low-traffic websites are often more challenging

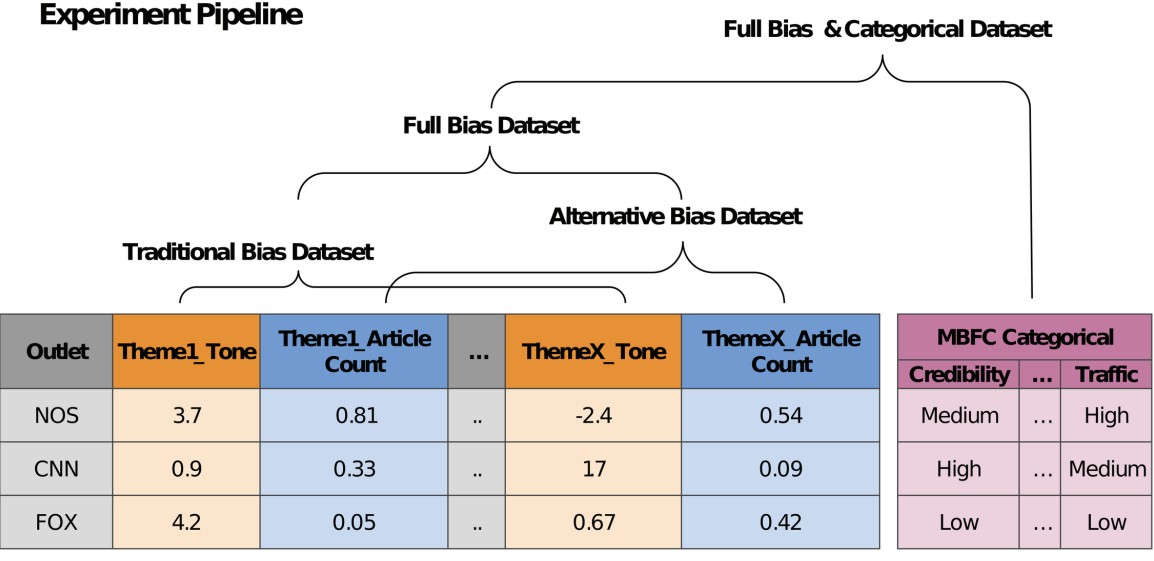

**Fig 3. Overview of experiments used to test models.** To test the impact of different bias-related data, models were trained on subsets of the data: traditional bias data (features related to tone, polarity, activity and self/group reference density); alternative bias data (features of word-, article-counts, image- or video presence); and the combination of all these features: full bias data. An additional experiment tested model performance on the full dataset when supplemented with categorical features from the MBFC data.

**Table 2. Model results per experiment.**

| Dataset | Model | TB Dataset | | AB Dataset | | FB Dataset | | FB&C Dataset | |
|---|---|---|---|---|---|---|---|---|---|
| | | Acc. | AUC | Acc. | AUC | Acc. | AUC | Acc. | AUC |
| **MBFC** | | | | | | | | | |
| | MBL | 0.454 | 0.500 | 0.454 | 0.500 | 0.454 | 0.500 | 0.454 | 0.500 |
| | SVM | 0.471 | 0.510 | 0.521 | 0.570 | 0.681 | 0.701 | 0.750 | 0.790 |
| | AdaB | 0.681 | 0.730 | 0.579 | 0.680 | **0.723** | **0.770** | 0.750 | 0.780 |
| | XGB | 0.622 | 0.700 | 0.546 | 0.640 | 0.655 | 0.720 | 0.648 | 0.690 |
| | NN | **0.689** | **0.730** | **0.630** | **0.710** | 0.664 | 0.750 | **0.765** | **0.813** |
| | Llama 3.1 | - | - | - | - | - | - | 0.531 | 0.622 |
| | GPT-4o mini | - | - | - | - | - | - | 0.555 | 0.627 |
| **PABS** | | | | | | | | | |
| | MBL | 0.408 | 0.500 | 0.408 | 0.500 | 0.408 | 0.500 | - | - |
| | SVM | 0.537 | 0.630 | 0.537 | 0.650 | 0.569 | **0.740** | - | - |
| | AdaB | **0.556** | **0.650** | **0.592** | 0.660 | 0.547 | 0.70 | - | - |
| | XGB | 0.444 | 0.580 | 0.531 | 0.640 | **0.579** | 0.660 | - | - |
| | NN | 0.479 | 0.620 | 0.582 | **0.700** | 0.527 | 0.680 | - | - |
| | Llama 3.1 | - | - | - | - | - | - | 0.495 | 0.632 |
| | GPT-4o mini | - | - | - | - | - | - | 0.505 | 0.623 |

The test set performance of models classifying political leaning of media web-domains under various experimental conditions: traditional bias features (TB Dataset; tone, polarity, activity- and self/group reference density), alternative bias features (AB Dataset; word- and article count, image- or video presence) and full bias features (FB Dataset). The last column to the right presents the performance of a model trained on the full data, supplemented with categorical features derived from MBFC's dataset (FB&C Dataset). Evaluation metrics are accuracy (Acc.) and AUC-score, and the best-performing model is highlighted in bold for each experiment. The tested models include a baseline model (MBL), a support vector machine (SVM), an AdaBoost model (AdaB), an XGBoost model (XGB) and a neural network (NN). Note that the LLM baselines using `Llama 3.1` and `GPT-4o mini` follow a slightly different implementation (see Appendix E), and are displayed under FB&C merely for the sake of brevity.

**Table 3. Output examples. Examples of domains with corresponding predictions and ground truths. Predictions were made using the best performing NN model.**

| Domain | Ground Truth | Prediction |
|---|---|---|
| investmentwatchblog.com | right | right |
| irishtimes.com | left | left |
| wishtv.com | left-leaning | least biased |
| 12news.com | least biased | least biased |
| khou.tv | least biased | least biased |
| bicesteradviser.com | least biased | least biased |
| dailyprogress.com | least biased | least biased |
| dailysignal.com | right | right-leaning |
| israelnationalnews.com | right-leaning | left-leaning |
| 12news.com | least biased | least biased |
| heraldpalladium.com | right-leaning | right-leaning |

when trying to determine political bias, but therefore also of central importance. Readers are expectedly aware of the political orientation of popular news sources like Fox News or The Guardian, and can therefore anticipate the slant of the information. The main difficulties arise when a reader encounters a lesser-known source, where the potential bias is unknown. This can be polarising in spheres where information is uncertain and moves at a rapid pace; as tends to be the case online, and especially on social media. A recent well-known case of this is the rebranding of Twitter to X, and the subsequent shift in tonality and bias. As such, our model's increased performance on lower-traffic websites is highly encouraging for dealing with unfamiliar sources' biases.

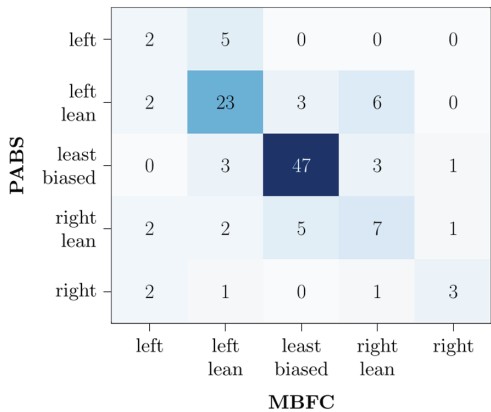

**(a)** Outlet bias classifications by the Neural Network for the traditional bias experiment.

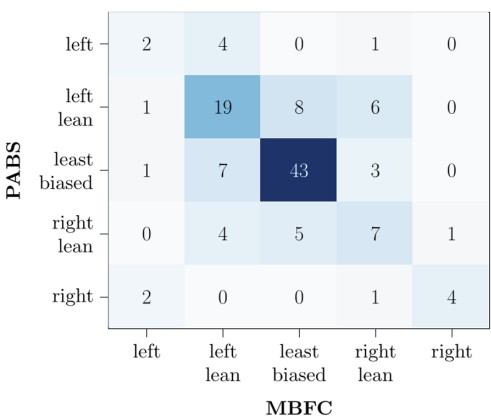

**(b)** Outlet bias classifications by the Neural Network for the alternative bias experiment.

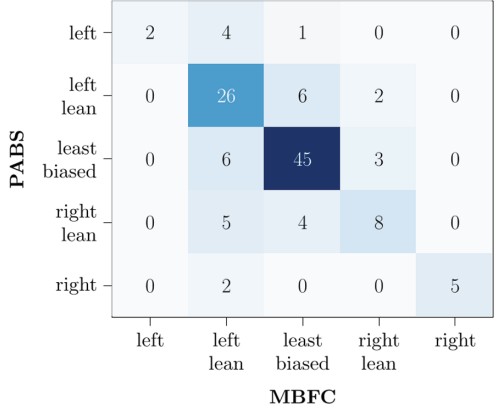

**(c)** Outlet bias classifications by the Neural Network for the full bias experiment.

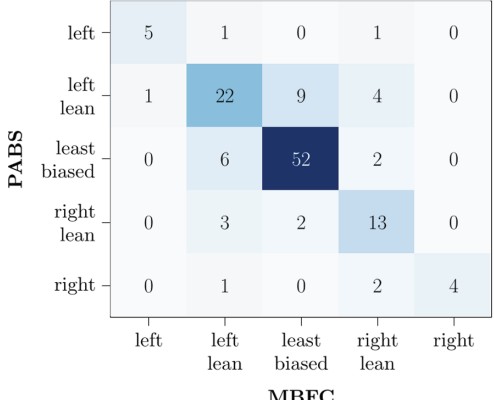

**(d)** Outlet bias classifications by the Neural Network for the full and categorical bias experiment.

**Fig 4. Confusion matrices.** Confusion matrices of the predictions by the best performing models per task.

## Model explanations

SHAP decision plots can be made for any given web-domain, so some representative examples were selected for visualisation. The results can be found in Figs 5–9 (due to limitations in the SHAP library, these pertain the comparably performing SVM model). Generally, the categorical features from MBFC frequently appear in the top most important features, with the exception of the Press Freedom Index. Geographical location also structurally appears as an informative feature in the provided examples. It is unclear what precisely about the coordinates influences the model. As the they refer to countries, rather than regions, this may reflect an approximate correlation with democracies and autocratic regimes. However, this should reasonably also be reflected in the Press Freedom Index, which seems to have been disregarded.

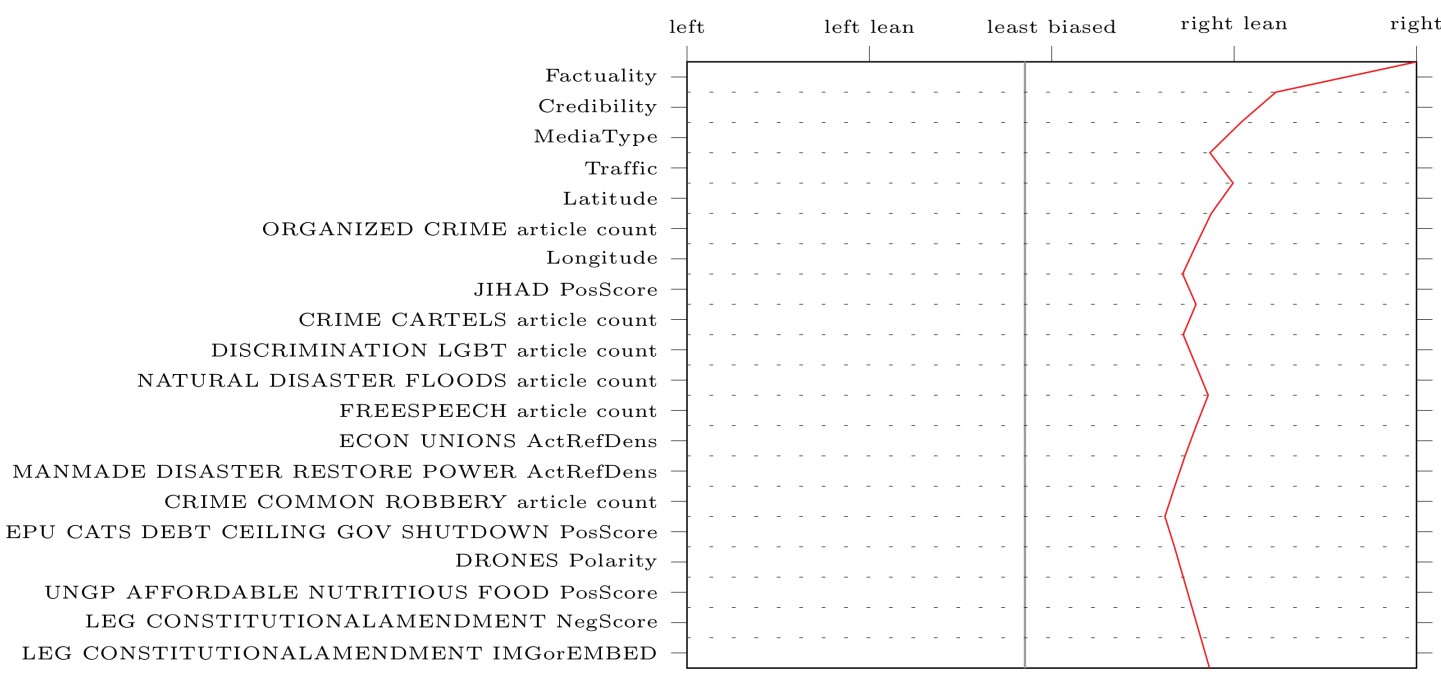

**Fig 5. Decision plot of Breitbart, a right-wing political news source.** The twenty most influential features are plotted in descending order. The range at the top of the graph represents the political bias labels as predicted by the model.

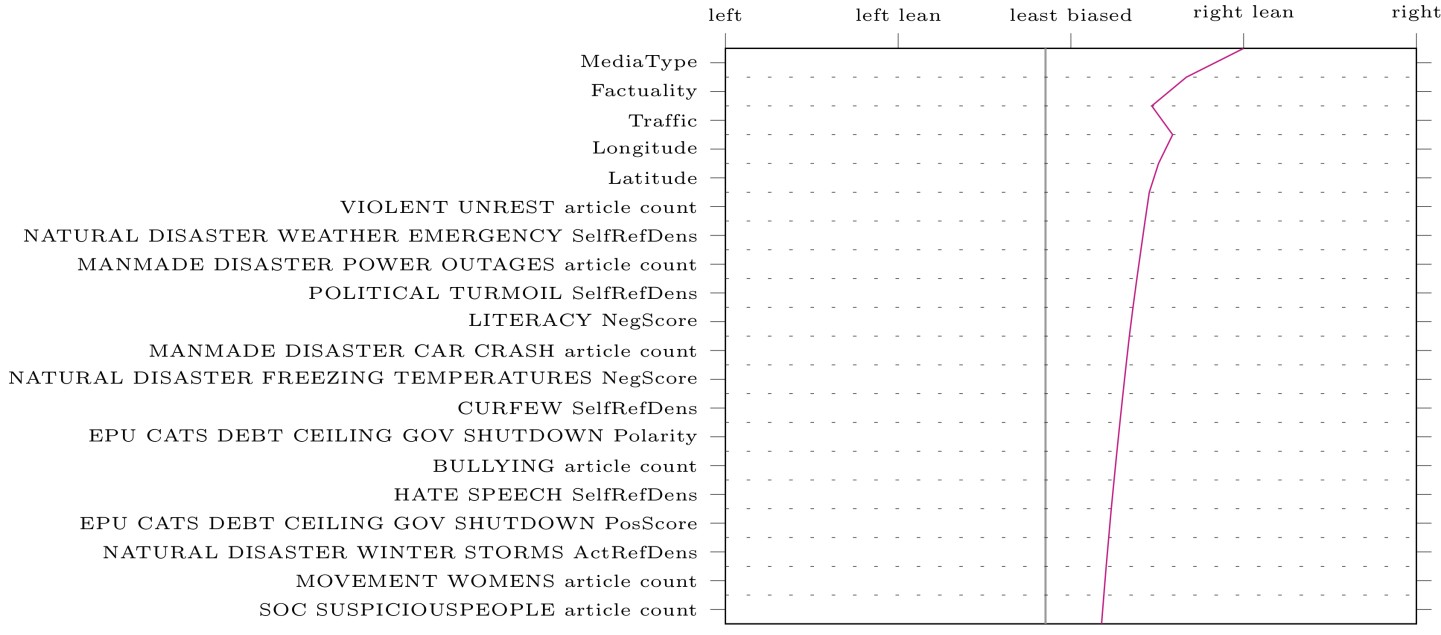

**Fig 6. Decision plot of Forbes, a right-leaning political news source.**

Aside from the categorical features, polarising themes previously highlighted in the literature are also prevalent (e.g., inequality, environmental issues, election fraud, firearm ownership, and social movements). Interestingly, however, the model also picks up on themes not appearing in earlier research (e.g., natural disasters).

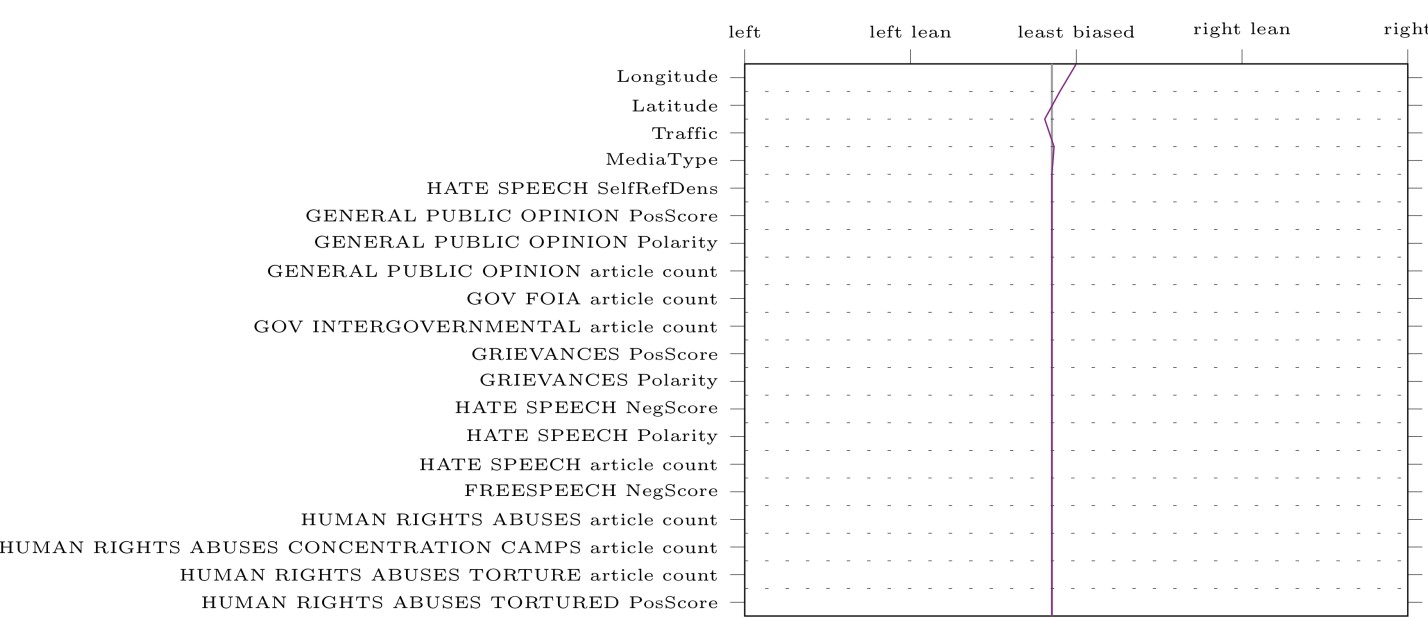

**Fig 7. Decision plot of the Economist, a centre-leaning political news source.**

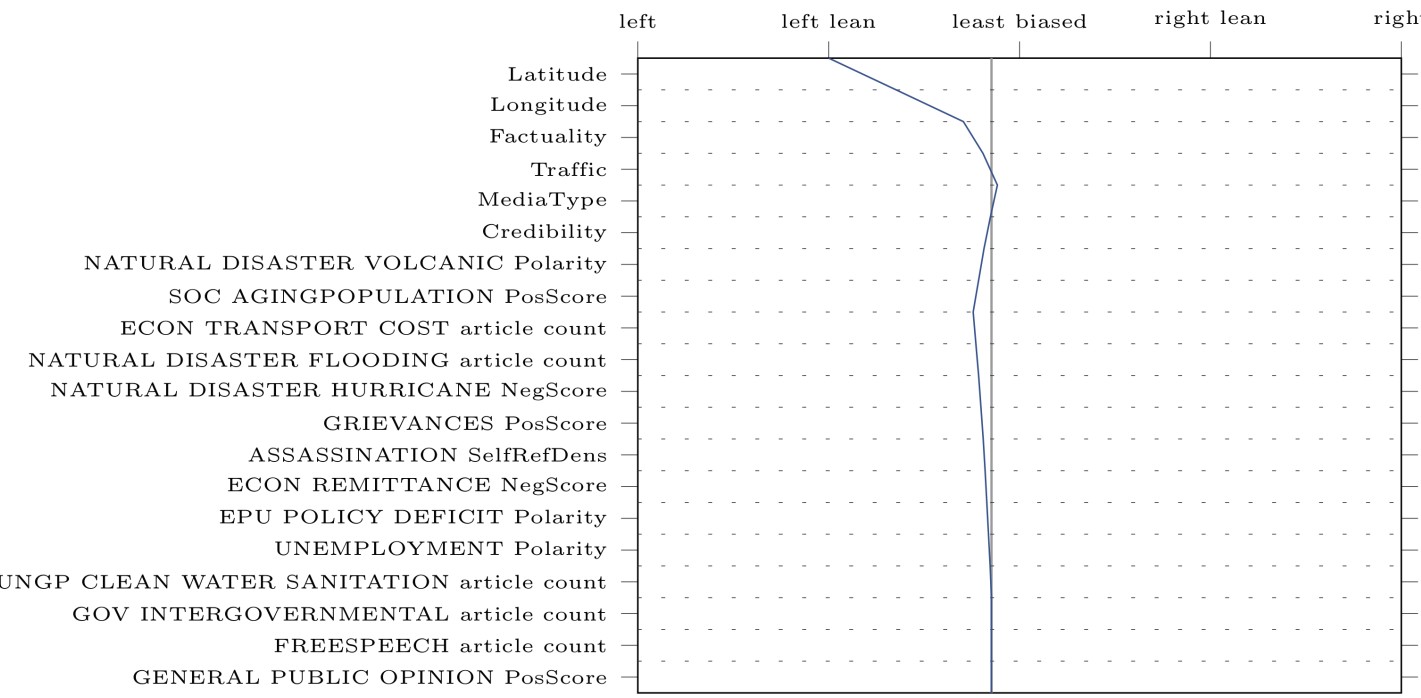

**Fig 8. Decision plot of the Guardian, a left-leaning political news source.**

Regarding the question of whether different types of bias impact model performance, article count features are often shown to be informative. This confirms the suggestion that alternative bias features could be more informative than previously credited: coverage bias, as represented by the number of articles published per theme, can be recognised and used to inform

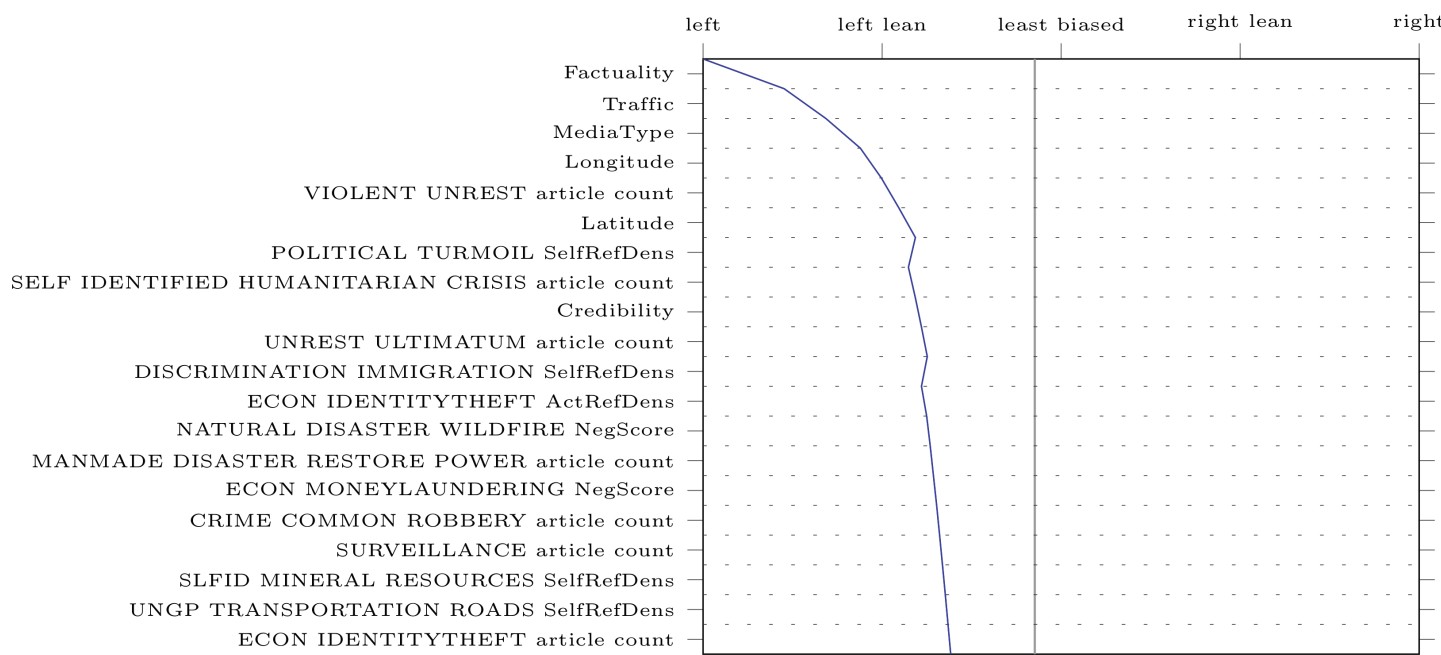

**Fig 9. Decision plot of CNN, a left-wing political news source.**

classifications. However, it is worth noting that other forms of alternative bias (such as word count or image presence) rarely appear in our decision plots, whereas traditional bias features are consistently informative.

Aside from gathering insight into how bias may manifest, SHAP can also be used to analyse misclassifications by the model (Fig 10). For example, the domain `theconservative-treehouse.com` was labelled as left-leaning despite actually being right-wing. Using the SHAP decision plot, we can see that the model was drastically influenced by the number of articles related to hate crime, causing it to output a left-centre label. Examining the dataset reveals that this domain has a high article count for this theme, and that this likely resulted in the misclassification. As such, SHAP plots can be helpful for analysing errors as well as understanding bias.

## Post-hoc analysis: Ground truth label comparisons

In light of the difference in performance between the models trained on PABS or MBFC data, we conducted additional analyses to examine this closer. A quick comparison revealed that there is a sizeable mismatch between labels. For all web-domains present in both PABS and MBFC's data, 46% of ratings agree with each other, and the AUC score is 69%. Fig 11 displays a confusion matrix to compare prediction errors, showing that neighbouring labels tend to be misclassified. Notably, there are some more significant disagreements; e.g., 32 left-wing web-domains are classified as "least biased" by PABS, and similarly for 14 right-wing web-domains. These larger gaps in labelling are problematic and raise an important issue regarding the validity of bias ratings in general, but especially in the disparity of results between computationally determined results and human-made labels. However, this particular labelling task is complex and prone to such disagreement even when based on human annotators. For example, MBFC and another bias rating website using human annotators, AllSides [65], show a slightly greater

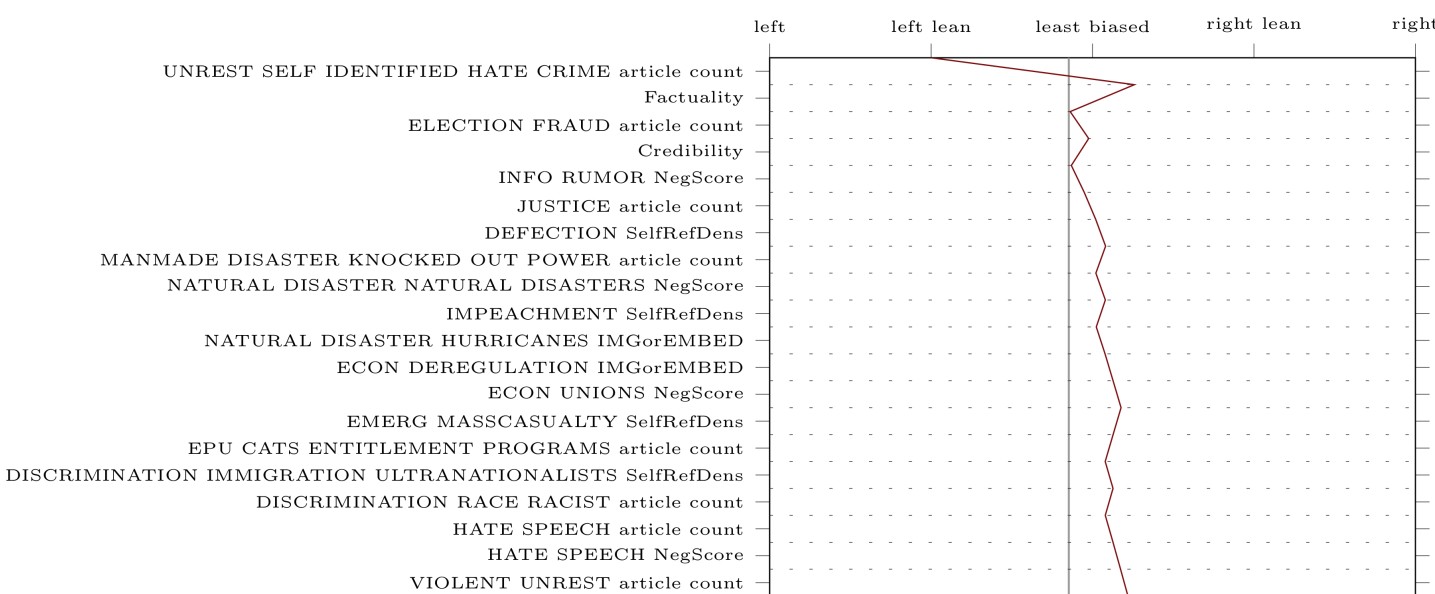

**Fig 10. SHAP decision plots of misclassified web-domain.** Example of a misclassified web-domain, `theconservativetreehouse.com`, which is a right-wing domain that was falsely classified as left-leaning by the model.

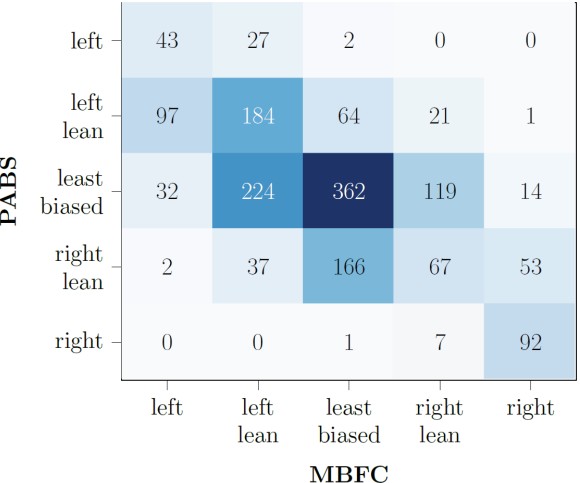

**Fig 11. PABS and MBFC label agreement.** A confusion matrix comparison of MBFC labels with those of PABS.

degree of agreement between themselves (57% of 293 web-domains in common agree, AUC score of 74%). Given that this is still quite a low degree of agreement, it demonstrates that the computationally determined labels are only slightly less reliable compared to the realistically achievable upper bound set by human annotations, while retaining the benefits of speed and efficiency.

## Discussion

The current work sought to automatically classify the political bias of news outlets, with a particular focus on scalability, minimal human intervention, and transparency. As GDELT

includes data from 1979 until the present, our approach allows for automatic labelling at any point in time, anywhere. Furthermore, it addresses limitations of previous research, such as the focus on constrained topics or types of bias [4,18,19,22,26]. Altogether, this allows for a much broader applicability than any previous research (to the best of our current knowledge), and is a fast and cost-effective method to recurrently obtain political bias estimates. Any user may employ our approach to analyse any given online website indexed by GDELT.

The leading question of this work was whether one can infer the political leaning of news web-domains based on GDELT data. As suggested by the results, this was indeed relatively successful. The highest performing model (a neural network) achieved an accuracy of 75% and an AUC score of 81%, compared to a 45% and 50% AUC score for the baseline model. This is comparable to previous work [19,22], though there is still room for improvement when compared to some of the more specific applications of media bias classification (though such implementations do not compare to ours in terms of scope [18]). Nevertheless, the fact that this approach could achieve such performance without using custom, optimized language models, but instead only using GDELT's relatively basic features is highly promising. Our results demonstrate the efficacy and value of our approach: even with a standard set of models and experiments, we achieved commendable performance.

Interestingly, despite the impressive capabilities usually ascribed to LLM models, the `Llama 3.1` and `GPT-4o mini` baselines performed poorly, assigning the "least biased" label to most items. These results mirror issues mentioned in previous work [27,28,31,32,35–37]. Our results are perhaps not surprising, as many of the news web-domains are not very well-known and therefore unlikely to have been sufficiently represented in the training data. In light of this, we would like to emphasize that our NN model performs best for outlets with low- or medium-traffic, as revealed during our error analysis, which sets it in an ideal position to detect bias for unfamiliar sources. This is crucial in the opaque and rapidly evolving online information sphere.

Another issue our work aimed to address was the focus on narrow types of bias in previous studies [4,18,19,22,26]. We compared the performance of models trained on the more commonly studied forms of bias related to word choice and general tone of articles (traditional bias dataset) to models trained on features related to under-explored patterns of bias (alternative bias dataset). The inclusion of alternative forms of bias improves performance, demonstrating that automatic bias detection benefits from expanding its focus. Indeed, features such as article counts per theme (a proxy for coverage bias) were particularly informative, as is apparent in Figs 5-9.

In addition to these advantages of the current approach, transparency and explainability of results were also central to our work. SHAP was used to provide detailed explanations of feature impact for any web-domain of interest. As an example, Fig 5 shows how Breitbart, a politically right-wing outlet, was accurately labelled thanks to features related to crime (including themes like cartels, kidnapping, black markets, organized crime, and robbery). The feature values can be interpreted to make sense of the result: for instance, many articles discuss crime cartels (0.545, where the maximum possible value is 1 due to the scaling of the data) and that the articles about black markets will tend to have a negative tone (0.409). Thus, one can interpret these results as meaning that Breitbart's focus on crime is indicative of right-wing bias, which is more intuitive and informative than a simple classification result or model-level feature importances.

Furthermore, the SHAP decision plots also lend credence to previous results. For example, alternative forms of bias such as article counts tend to be in the top twenty most impactful features in Figs 5-9. Nevertheless, the plots also raise some questions. Some themes are part

of the "usual suspects" of politically polarising themes in previous work (inequality, environment, social movements, firearms, and election fraud). Other themes, however, are more difficult to explain. For instance, features related to exhumation, sanitation and natural or man-made disasters of varying kinds are also included in the top twenty features, though these are more difficult to fully interpret.

Aside from this, we should also highlight that models performed better when trained on human-made labels [45] compared to computationally derived ones [25]. A potential reason for the drop in performance is that these labels are inherently an approximation, and therefore add uncertainty for the model. This is particularly detrimental when such approximations compound. Robertson et al. [25] used Twitter sharing patterns of registered voters to attribute a score to each website when creating the PABS labels we used as ground truth. However, as they themselves note, this assumes that people only share articles that agree with their own political opinions, which is not always the case (they do perform various tests for validity with other existing bias labels that suggest their results are an adequate replacement, however). Nevertheless, the present study's results suggest that models trained on such computationally determined labels can still be used to some extent, should manual labels be unavailable. Furthermore, it is worth noting the degree to which the models generalize, indicating that while manual labour was initially required, we can now partially rely on these models for subsequent analyses even without human-made labels.

Altogether, bias detection following an approach similar to ours could hopefully be more informative for the public, offering a transparent examination of overall web-domain behaviour. This can be done in a cost-effective and recurrent way, allowing for systematic estimates of political bias in the online news media environment. This might contribute to citizens' ability to make decisions in an informed manner about various topics important in the current political climate [66].

## Limitations and future considerations

Despite addressing many drawbacks of previous research, such as the manual annotations, limited applicability, and focus on narrow forms of bias, there are remaining limitations of the current approach as well.

Firstly, the best-performing model was trained on GDELT and supplemented with categorical features from MBFC. These categorical features are, however, only available for a subset of web-domains present in GDELT, meaning that this particular model is not applicable to all web-domains. Nevertheless, other models trained only on GDELT data achieved comparable performance, so it is possible to get accurate predictions and SHAP explanations for any GDELT website with a minimal drop in accuracy.

Secondly, better features representing the various forms of bias could be constructed in the future. Picture and explanation bias was, for example, only indirectly examined here, as the current approach only accounted for the presence or absence of images. Ideally, the actual content of the images would be included, as has been done in previous studies using GDELT [39]. Additionally, some forms of bias were excluded from this analysis despite potential relevance (e.g. placement bias).

Furthermore, it was noted during post-hoc analysis that the ground truth labels display remarkable disagreement with each other. This raises questions regarding what can be considered acceptable ground truths, as even expert labels tend to disagree with each other. Future work might want to consider using other labels of political bias, as the bipartisan scoring does not necessarily lend itself well to all global political systems. Indeed, it has been noted that what is considered left-leaning in one country would not be so in another one [3]. As

such, future applications of our approach should remain aware that the labels we use may not perfectly transfer when considering another country. Helpfully, however, MBFC provides an overview of criteria for how their bias labels were determined (for example, the outlet's stance on taxation, abortion, or the climate), which can be used to evaluate whether the scale is applicable for any particular use case [44]. Given previous research on the persistence of the left-right political divide [1] and the pervasiveness of US political structures in its media and, importantly, its social media, we expect that these ground truth labels will be appropriate in the majority of, at the very least, Western countries. Should future work wish to forgo the left-right political divide for another kind of distinction, our approach nevertheless remains helpful, since the model can quite simply be retrained on the same data but using a different set of ground truth labels. This might enhance the applicability of our approach to countries that do not neatly follow the left-right political spectrum, while retaining the benefits of a systematic and in-depth analysis of political bias in news. Alternatively, future work may forgo using labels altogether, opting for unsupervised models instead. Given that there are no perfectly unbiased benchmarks, this may be a preferable approach depending on the context. Future work may also examine the possibility of using GDELT's data to examine an outlet's overall stance on a particular theme. This could provide a robust and extensive perspective on news outlets, and show potentially unexpected biases.

The rise in popularity of LLM-based methods also presents a promising avenue of research, despite the challenges mentioned in our section on related work. Given the unprecedented potential for nuanced model output explanations that LLMs offer, they certainly merit further study of whether they can provide the nuance commonly reserved to expert-based approaches.

Lastly, our use of SHAP is explorative; future work in online bias might focus on including more detailed information, such as relevant excerpts of articles, to concretely provide insight into model predictions. This might give insight into about why some of the more surprising themes were deemed to be informative, such as waterways, for instance. Generally, it can help inform the field of themes that are not usually considered in online bias research.

## Conclusion

The current work proposes an approach to classify news outlet political bias. Crucially, we intended to expand the scope beyond what had previously been done by ensuring global coverage and by focusing on multiple forms of bias. Our results indicate that the method indeed provides a fully automatic and scalable approach to detecting news bias, and that enlarging the focus to multiple forms of bias could help the field advance. Finally, the SHAP explanations allow for interpretation of why a particular web-domain is considered politically biased and show which topics and behaviours influence the classification. Interestingly, many themes commonly considered divisive reappear, but some informative features have not been previously considered in the literature. This may help address gaps or future avenues in current research.

All in all, the current work extends existing research to be more widely applicable and informative for the field. The increased transparency may be helpful for adequately informing the public about its news consumption, as well as providing more insight into the underlying mechanisms of bias to a more granular extent than what is traditionally attempted by computational methods. Considering the immense impact of news on global political climates, our hope is that increased understanding and trustworthiness of media might contribute to a better-informed society and a healthier political environment.

## Supporting information

**S1 Appendix. MBFC features.** This appendix describes in more detail the features provided by MBFC. For each news web-domain, they provide a set of data points of interest.
(PDF)

**S2 Appendix. Details of implementation and data pre-processing.** This appendix describes in the implementation and other details related to the data pre-processing.
(PDF)

**S3 Appendix. Grid search model parameters.** The appendix details the hyperparameters used for grid search when optimizing the various models.
(PDF)

**S4 Appendix. Neural network architecture.** This section notes the architecture of the PyTorch neural network for all experiments.
(PDF)

**S5 Appendix. Large language model baseline.** This section notes the implementation details of the LLM baseline.
(PDF)

**S6 Appendix. Error analysis.** This section details some error analysis per political leaning label and the website traffic.
(PDF)

**S7 Appendix. Example of a GDELT news item.** Shows an excerpt from a news story as it is shown on GDELT.
(PDF)

## Author contributions

**Conceptualization:** Ronja Rönnback, Chris Emmery, Henry Brighton.

**Data curation:** Ronja Rönnback, Chris Emmery.

**Formal analysis:** Ronja Rönnback.

**Investigation:** Ronja Rönnback.

**Methodology:** Ronja Rönnback, Chris Emmery, Henry Brighton.

**Project administration:** Ronja Rönnback.

**Software:** Ronja Rönnback, Chris Emmery.

**Supervision:** Chris Emmery, Henry Brighton.

**Validation:** Ronja Rönnback, Chris Emmery, Henry Brighton.

**Visualization:** Ronja Rönnback, Henry Brighton.

**Writing – original draft:** Ronja Rönnback.

**Writing – review & editing:** Ronja Rönnback, Chris Emmery, Henry Brighton.

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
