## [Decision Letter · Decision Letter 0]

5 Jan 2025

PONE-D-24-54904Automatic large-scale political bias detection of news outletsPLOS ONE

Dear Dr. Thelen-Ronnback,

Thank you for submitting your manuscript to PLOS ONE. After careful consideration, we feel that it has merit but does not fully meet PLOS ONE’s publication criteria as it currently stands. Therefore, we invite you to submit a revised version of the manuscript that addresses the points raised during the review process.

We look forward to receiving your revised manuscript.

Kind regards,

Shady Elbassuoni, PhD

Academic Editor

PLOS ONE

Journal Requirements:

Reviewers' comments:

Reviewer's Responses to Questions

**Comments to the Author**

1. Is the manuscript technically sound, and do the data support the conclusions?

Reviewer #1: Yes

Reviewer #2: Yes

Reviewer #3: Yes

2. Has the statistical analysis been performed appropriately and rigorously? 

Reviewer #1: Yes

Reviewer #2: Yes

Reviewer #3: Yes

3. Have the authors made all data underlying the findings in their manuscript fully available?

Reviewer #1: No

Reviewer #2: No

Reviewer #3: No

4. Is the manuscript presented in an intelligible fashion and written in standard English?

Reviewer #1: Yes

Reviewer #2: Yes

Reviewer #3: Yes

5. Review Comments to the Author

Reviewer #1: Report on “Automatic large-scale political bias detection of news outlets”

The paper presents a new way of analyzing news outlets with regards to a potential political bias in the presentation of news. The authors measure this bias as the deviation from a centered position along the typical left-right paradigm according to numerous parameters, which include not only socio-economic content, but also media type or geographical information on the outlets. The authors are able to position the analyzed news outlets by means of an ML-trained algorithm appropriately. Appropriately here means, in relation to two benchmarks.

The authors’ comparison with these benchmarks reveals that the algorithm does not perform significantly worse than manual or computer-aided classification. This means that the ML-algorithms are able to show biases quickly, cheaply and on a broad scale, but at the cost of lower accuracy. And as they apply the algorithm to the GDELT-database a near real-time computation is facilitated. The new method, however, does not achieve an outperformance of the traditional measures, but given that ML allows rapid learning and improvement, the value of the paper is its new methodology, the application on the GDELT database. In a world where news outlets change quickly, the way news content is distributed evolves, it might be more important to receive results quickly for many outlets than to have the best answer late and for only a smaller number of outlets. What is more, the accuracy can likely be improved with next level ML-models.

The topic is relevant. Biases in media can be a problem, as a well-informed audience is the base for a well-functioning democracy and efficient markets, too. Receiving unexpectedly biased information can impact the decision-making processes within a democracy and lead to inefficient or even failing markets. Having a better understanding of existing biases can then help to maneuver in a world full of uncertainties.

The paper has major and some minor limitations, and the authors are aware of most of them. For example, the left-right-paradigm (bipartisanship) might be a good approximation of political reality in the US today, but this paradigm has been challenged at least since the 1930s and surely by the materialism/postmaterialism cleavage introduced by Ronald Inglehart in the 1960s as well as by the evolution of new cleavages like the ecological orientation. However, as there is no common sense about the right number and structure of cleavages in a society, the left-right-paradigm is still the least common denominator for political analysis today. The bigger obstacle is to my view is therefore not the selection of the left-right paradigm, but the implicit multidimensionality and instability of this paradigm. While mostly a market-oriented economic policy is viewed as a “right-wing” economic policy, the very right-wing, fascist views often favor strong governments and less economic freedom. This can only be overcome by providing the complete analysis and not one single aggregated bias-score. As second problem of the left-right paradigm is that the dimensions of what constitutes a right-wing position can be subject to change. In many northern European countries religious questions do not constitute a typical conservative view anymore, this is different in the US. Thus, the trained algorithm must ensure that these regional differences are captured in the case of multi-country analyses. It would be good if the authors were giving more guidance if and how their algorithms are able to deal with these differences.

Additionally, it would be important to motivate the topic better. Biased information might not be a problem for democracy, if the biases of the outlet are well-known. Breitbart for example is known to be a right-wing outlet. The Guardian is known to be left-wing. The value of measuring biases is only for cases of unexpected bias. The authors do not seem to distinguish between expected and unexpected biases. The known biases can be considered an element of free speech and different attitudes and values of people. The unexpected biases can be rather considered an element of deceiving the reader (willingly or unwillingly). Thus, biases are a problem in a rapidly changing news outlet environment (e.g. when twitter turned into X) and in case of opaque positions. It might also be worth elaborating that the benchmark itself can be a moving target. What is centrist today can be easily be considered left-wing 50 years from now. What is considered socialist in the US might be considered centrist in Scandinavia. These possible changes are a good argument for running frequent and cost-efficient analyses, i.e. ML-driven analyses.

The authors also claim that the relevance for biases lies not only in what is covered, but also in what is not covered. I subscribe to that, but had difficulties finding, where this new element is covered in the algorithm.

The authors rightly point to a crucial limitation, that an LLM which is trained on a biased text will mirror this bias. This can be dealt with by analyzing the broadest possible database (e.g. GDELT); however, even this database might be subject to constraints. Are extreme positions covered? Does the analyzed text universe mirror the population’s views adequately and representatively? In a world with rising concentration rates in media, it might be doubted that the possible bias in traditional media outlets that rises due to the concentration process is sufficiently compensated by free social media – when social media is partly curated and/or assumed to amplify more extreme positions.

This leads directly to the key limitation of the paper: The authors measure the differences between their results and two benchmarks. But provided that there is no guarantee that these benchmarks are unbiased, who can really say, which computation gives the better estimate of a bias? While there won’t be simple answer to this question, the authors should discuss this issue, as they want to measure biases rather than model existing estimates for biases.

I advise also that the authors elaborate on some of their results (Fig. 5 and Fig. 6) in more detail. While the overall classification seems plausible (which implies that for the given examples the overall results to not yield much of additional information; the measured bias mirrors my own perceived bias for the respective outlets). Some of the parameters that were derived call for a discussion: why are geographic information so important? Does this reflect national or rather regional differences? How does media type lead to a bias?

Some of the authors’ line of argument is based on the assumption that people “predominantly share links to domains they agreed with”. While this is plausible, it is worth pointing out, that this can be questioned, at least in comparison to traditional outlets (see e.g. Seth Stephens-Davidowitz (2017). everybody lies, Bloomsbury). For readers it is often evident, when a link to another domain is meant as a recommendation or is combined with an aggressive, hateful or just unfriendly connotation; a computational system can easily miss these cynical or even hateful connections and might count them as support. The authors should elaborate whether these link-based classifications are able to distinguish between these different forms of links – or why it does not matter (e.g. maybe one is significantly more important).

The authors need to go through the document and check all links to figures, particularly the references to Figure 6.

Concluding, the paper has three important strengths: 1) application of GDELT-data by means of ML-models to political biases. This is a new way of measuring biases automatedly and can be the introduction of large-scale classifications 2) the Shap values allow to show the main drivers of the results. This additional information is important for understanding biases 3) the paper shows that the LM models can better explain human labeling than computer-aided labeling. To my view, the authors should highlight this finding more, as it might be interpreted that the computer labeling creates a blur itself.

These strengths come on the back of limitations. Some cannot be dealt with easily (as the authors know), others can be addressed: 1) motivate better, why a better and faster understanding of news biases is an important topic and what kind of biases really need more transparency (known versus unknown biases). 2) how can we enhance the model for countries that do not follow bipartisanship as nicely as the US? 3) how can this be scaled up? By whom and how?

Reviewer #2: This is an interesting and well-written article looking at the possibilities of detecting features of journalistic content using computational methods. In my view, it uses a sound methodology and can be a good fit for the journal. However, before it can be considered for publication, I would suggest the author(s) to make a selection of revisions to strengthen the manuscript.

My first and major concern regards the use of the term bias throughout the article. As the authors note themselves, it is a complex concept which can be used rather differently. However, there is no clear definition of bias used in the article - the authors discuss its different forms, but the overarching concept remains unclear. As a result, I am not certain at all that the concept of bias is actually helpful for the study as it, in its core, does not actually look at bias, but on certain content features. By calling presence/absence of these features biases, the authors in my view dilute the concept due to two reasons: 1) first, bias usually refers to systematic skewness in the presence of certain features; the paper, however, does not look at systematic/non-systematic aspects of these features' presence; 2) if we assume that all these features indicate bias, then my question is what would be unbiased way of selecting events or presenting pictures in journalistic materials? If essentially everything is a form of bias, then what kind of use does the concept have?

Besides it, I have a few additional points which I would suggest the authors to consider:

-While LLMs do offer a promising direction of automated content enrichment, it is important to consider how stable are their assessments and how prone they may be to stochasticity. I think it will be beneficial to briefly note this limitation of them being applied for content analysis (for examples, see Motoki, F., Pinho Neto, V., & Rodrigues, V. (2024). More human than human: measuring ChatGPT political bias. Public Choice, 198(1), 3-23; Makhortykh, M., Sydorova, M., Baghumyan, A., Vziatysheva, V., & Kuznetsova, E. (2024). Stochastic lies: How LLM-powered chatbots deal with Russian disinformation about the war in Ukraine. Harvard Kennedy School Misinformation Review.)

-It will be advantageous to briefly discuss how generalizable is approach for bias/feature detection for other types of content (i.e. beyond journalistic materials). Would different set of features be more relevant there? Will the same approach be directly transferable?

-It will be useful to briefly discuss other studies which use GDELT and MBFC datasets for research purposes to demonstrate that there are reliable tools for the tasks discussed in the article.

-A brief discussion of possible alternatives for SHAP can be useful in my view.

Reviewer #3: The article manuscript Rönnback et al. is concerned with the machine-learning based automatic news bias classification of online news Web sites at scale; the authors classify news Web sites at the outlet level according to the polarity of political bias of the writing they host. To this end, the authors train a range of supervised machine learning models with a mix of three pre-existing datasets collected by others (GDELT).

The paper is well-enough motivated regarding the goal and the paper's topic importance for sustaining democracy. The purpose of the paper is to facilitate the study of news bias on the Web at the "news outlet" (Internet domain) level.

The paper is clearly written for the most part (particular omissions and points of lack of clarity will be pointed out as we go along).

The Related Work section contains references that are all on-point. However, the coverage of the relevant literature is not detailed enough for a peer reviewed journal article in such an active area (its level of substance looks more in line with what one would expect in a conference paper). Most importantly, the seminal KDD paper by Ye and Skiena (2019), who recently also applied bias detection at outlet-level on scale, which is arguably the mosts similar paper to the proposed article under review has NOT been included in the discussion of related work. One would expect that a comparison between the authors' plans and what Ye and Skiena did (their MediaRank report tracked 50,696 news sources from 68 countries) was a majort part of the Related Work section, and that beyond that, Ye and Skiena's data and method should be also used as a baseline if possible, or, alternatively, at least it should be discussed why this is not possible.

This omission means that the state of the art is not properly recpgnized, but as a consequence also the novelty of this manuscript suffers (some of this might be curable, see my suggestions below).

The paper offers some criticism of using LLMs on the task, all the while also praising the "opportunities" LLMs offer in vague terms; I read this as handwaiving for not using an LLM-based approach in the list of methods for comparison. The authors should make their partially interesting paper stronger by cutting vague language and by comparing their existing range of classifiers with a zero-shot GPT-4.0 and/or Llama experiment, which is easy to apply and fast, as no annotated data and no training time/money are required, only a prompt needs to be issued to the LLM. They should then compare the performance adn state advantages (rapid development, in < one hour, excellent quality to effort ratio) and disadvantages like energy consumption, potential cost for proprietary models, hardware requirements etc).

In order to get the paper publication-ready, the following six areas of issues should be at least addressed:

(a) The selection bias that results from

(i) the use of the GDELT database, and also the bias that results from

(ii) sampling of the pandemic year 2022, should be discussed.

(b) Add and describe the seminal paper Ye and Skiena (2019), Godbole, Srinivasaiah and Skiena (2007), Kulkarni et al. (2018) and other missing important and closely related pieces of prior work (e.g. Hamborg (2023); Menzner and Leidner (2024a,b), etc.). At least Compare Ye and Skiena's (2019) and your own results; better would be adding a baseline similar to their work in your comparison table.

(c) Error analysis: include more detailed qualitative error analysis with examples - what groups of errors have you encountered? In particular, could you investigate the root cause why the particular left-leaning and right-extremist sites were misclassified (which ones are affected)? Can you describe an experiment with which one could find out?

(d) Provide some sample input and output:

- In the Introduction, show an example GDELT biased news story or snippet thereof, to give the reader a sense of what the authors' data looks like.

- In the Results section, add a table or figure depicting some sample domains and where they land on the political spectrum, as classified by your best model.

(e) Experimentation:

- mandatory: add an experimental result with a zero-shot LLM as a comparison (using a SOTA and easily accessible model like either GPT-4 and/or Llama-3.1, for instance).

- optional: once you have completed the above experiment, an easy-to-do follow-up presents itself

using a stacked ensemble: append the LLM's output label to the feature vector of your own neural model.

(f) State whether the resulting data and code are available or not, and if so, include a hyperlink to the repository.

Once these points are diligently addressed, I would be supportive of publishing this paper.

Editorial Comments/Corrections/Suggestions

line 4 and throughout the manuscript: web -> Web (World Wide Web is a proper noun/name)

line 10, after "uncontroversial": add a citation to support this?

line 75: after TFIDF there is no citation, but after word2vec there is: fix this by citing Spärck Jones (1970) J. Doc.

line 103: "unique opportunity": 1. vague language, 2. not just article-level, also sentence-level 3. cite peer-reviewed example not just pre-prints, e.g. Menzner and Leidner (2024a,b)

line 116: "LLMs are ultimatelz less fitting.": quite a general statement without any support; also, if I was you, I would not say that given how early our experience with LLMs is. And it's a self-contradiction with line 102-3.

page 5, Table 1: explain what "active words" are. Even better, give at least one example for each row of this table in an extra column.

line 214: missing whitespace (indicated by "_") in "between Appendix C._All other models"

References: some bibliogr. information missing [27-29], [43-44], [15] has now been published at a peer reviewed conference - please update

References

• Hamborg (2023) Revealing Media Bias in News Articles NLP Techniques for Automated Frame Analysis

https://link.springer.com/book/10.1007/978-3-031-17693-7

• Menzner and Leidner (2024) "Experiments in News Bias Detection with Pre-trained Neural Transformers" Proc. ECIR, Glasgow, Springer Nature.

• Menzner and Leidner (2024) "Improved Models for Media Bias Detection and Subcategorization" Proc. NLDB, Turin, Springer Nature.

• Ye and Skiena (2019) "MediaRank: Computational Ranking of Online News Sources" Proc. KDD, Anchorage, AL, USA.

• Godbole, Srinivasaiah and Skiena (2007) "Large-Scale Sentiment Analysis for News and Blogs" Proc. International Conference on Weblogs and Social Media, 219–222.

• Kulkarni et al. (2018) "Multi- view Models for Political Ideology Detection of News Articles" Proc.

EMNLP.

6. PLOS authors have the option to publish the peer review history of their article (what does this mean?). If published, this will include your full peer review and any attached files.

Reviewer #1: No

Reviewer #2: No

Reviewer #3: No

---

## [Author Response · Author response to Decision Letter 1]

10 Feb 2025

This document details our responses per point raised by reviewers. We want to firstly thank reviewers for their insightful comments and suggestions, and we have attempted to address each point in the revised version of the paper.

We would firstly like to thank the editor for their assistance on this submission, and have made modifications relating to the feedback. Namely, we have made a minimal dataset available necessary for future replication purposes at a public repository (https://github.com/rtronnback/automatic_news_monitoring_with_GDELT ). Similarly, we have verified that all pictures pass through the PACE tool and uploaded the PACE generated figures. We also ensured that figures and tables were all included immediately after they were mentioned, instead of LaTeX’s default placements, and corrected file names.

We thank Reviewer 1 for their useful feedback, we appreciated the reviewer’s awareness of the broader context and the considerations for making our approach more applicable for wider use, particularly regarding systems that do not follow the Left-Right political divide. We think the paper has greatly improved thanks to their suggestions. In response to the comments, we have added guidance on the use of our approach in a cross-country setting (and refer to previous work that explores criticisms of the left-right political divide, see lines 499-500). Namely, users should be aware that the left-right political divide our project is based on uses the bias labels made by from Media Bias Fact Check. They provide an overview of contrasting left-right positions that is informative for deciding whether this political scale will be fully applicable to a country of interest. Given the persistence of the left-right divide, as per (Le Gall & Berton, 2013), it can be assumed that these ground truth labels are helpful in the majority of, at the very least, Western countries. We advise people interested in other divides to use their own GT labels, as our approach will still be applicable (lines 508-519). As we make our code fully available on GitHub, our method is easily accessible for alternative approaches.

We have, as suggested, added a specification about unexpected biases (lines 22-23), and the benefits of cost-effective and recurrent analyses (lines 405-408; 472-477). Regarding the relevance of what is covered versus not covered to determine bias, we have clarified this in the Methods section (lines 213-222).

We have added nuance related to the fact that no benchmarks are perfect, and suggested alternative approaches (lines 245-249; 508-519). We have also highlighted the finding that models perform better on human-made labels, and that this might be due to computational labels adding noise that complicate downstream analyses (lines 457-461).

We have elaborated on results in Figures 5 and 6 (now split into individual figures) and added a new example of a misclassified result. We have also fixed references to Figure 6.

Regarding the point of how this approach can be scaled up, we have highlighted the benefits of using GDELT, as this allows for global application of our approach (lines 472-477; 497-519).

We thank Reviewer 2 for their insightful suggestions regarding how to strengthen our paper, especially for pointing out some aspects which were not sufficiently clear in the original version. In response to the comments, we’d like to reaffirm that our paper does examine skewness in the chosen GDELT features (since the systematic presence of absence of features are linked to a political bias label by the ML models), and the revisions have hopefully clarified this. Additionally, we have added a definition of bias from the literature that better reflects the broad and nuanced nature of the subject, while improving clarity for readers (lines 61-63). We have added the recommended papers regarding LLM stochasticity, and are particularly appreciative of the recommendation of the work by Makhortykh et al. (2024).

Regarding the generalizability of our approach beyond journalistic materials, we have specified in the paper that our current approach extends to what is covered by GDELT. As GDELT is a uniquely detailed source, any project looking to extend a similar analysis beyond news outlets would have to first compile a dataset on the other topic in question using a similar structure as GDELT. However, GDELT is quite extensive – there are trillions of datapoints in their database, and a single year totals 2.5TB of data before aggregation. Therefore, we hope that it will cover most use cases that involve any online content (see lines 189-200; 494-519).

We have specified when a paper we mention has also used GDELT or Media Bias Fact Check, and added examples of existing work that relies on it to demonstrate that they are reliable tools (lines 190-192; 249-250; 491-492). We have also mentioned some common SHAP alternative methods used for model explainability (lines 287-289).

We thank Reviewer 3 for the clarity, depth, and constructiveness of their comments. The feedback has been greatly helpful during our revision of the paper. In response to the comments, regarding point (a) and the selection bias resulting from using GDELT and a pandemic-year, we have elaborated that GDELT is currently the single largest database covering news on a global scale, and which we are confident mitigates selection bias, even despite the effects of the COVID-19 outbreak (lines 202-205).

For point (b), we have expanded upon the literature section as suggested, and would like to thank the reviewer for the recommendations of related papers. We have added the relevant papers (lines 111-115; 127-141; 145-158) as suggested by reviewer 3. Particularly the work by Ye and Skiena (2019) was examined and compared to our work, and some interesting differences were elaborated upon. It was however ultimately not added as a baseline. Despite the original paper using a sentiment score between -1 and 1, the published database transforms this to a bias rank, thus removing distinction of the left-right divide that out approach relies on. Due to this, we cannot compare the two approaches, since it provides only an estimate of the amplitude of bias and not its direction, as our work does. Therefore, comparison would necessitate a fundamental change for our approach that is not compatible with the current focus on a left-right political divide. Furthermore, few domains in our test set were to be found in the database, making comparison difficult. As mentioned above, however, future projects can use our code for such alternative ground truth labels, should that be the point of focus.

For point (c), we have added examples of errors with analysis as a way of interpreting the model’s behaviour, particularly for the traffic of the domain, and the classification label (see Appendix F). We have also added a demonstrative example of using SHAP to examine misclassifications (see Figure 10).

As per point (d) in suggested edits, we have added an example of the GDELT data before it is aggregated to Appendix G. Furthermore, the Results section now includes Table 3 which shows an overview of domains, their true labels, and the classifications by our best model.

Regarding point (e) and the addition of a zero-shot LLM experiment, we have taken the reviewer’s suggestions, refined the text (lines 142-165), and performed a baseline using Llama 3.1 and GPT4-io mini (see Table 2, and Appendix E). Given that the GDELT aggregate data used to train the other models is purely numerical, we opted for a naïve zero-shot baseline using the domain name, thereby relying on the LLM’s latent representations of the domains. We have, during other projects, experimented with using LLMs in this way with more elaborate prompts, but have to date only achieved middling results at best.

As per point (f), the code used for this analysis, as well as the training, validation and testing data, is available on Github (see footnote on page 8).

We once again wish to heartily thank the editors and reviewers for their valuable feedback that has allowed us to noticeably improve on the paper.

References:

Le Gall C, Berton R. Left-Right vs. traditional and new cleavages: testing durability of an old political category. In: Left and right : the great dichotomy revisted. Cambridge Scholars Publishing; 2013.

Makhortykh M, Sydorova M, Baghumyan A, Vziatysheva V, Kuznetsova E. Stochastic lies: How LLM-powered chatbots deal with Russian disinformation about the war in Ukraine. Harvard Kennedy School Misinformation Review;doi:10.37016/mr-2020-154.

Ye J, Skiena S. MediaRank: Computational Ranking of Online News Sources. In: Teredesai A, Kumar V, Li Y, Rosales R, Terzi E, Karypis G, editors. Proceedings of the 25th ACM SIGKDD International Conference on Knowledge Discovery & Data Mining, KDD 2019, Anchorage, AK, USA, August 4-8, 2019. ACM; 2019. p.2469–2477. Available from: ttps://doi.org/10.1145/3292500.3330709.

---

## [Decision Letter · Decision Letter 1]

7 Mar 2025

Automatic large-scale political bias detection of news outlets

PONE-D-24-54904R1

Dear Dr. Thelen-Ronnback,

We’re pleased to inform you that your manuscript has been judged scientifically suitable for publication and will be formally accepted for publication once it meets all outstanding technical requirements.

Kind regards,

Shady Elbassuoni, PhD

Academic Editor

PLOS ONE

Additional Editor Comments (optional):

Reviewers' comments:

Reviewer's Responses to Questions

**Comments to the Author**

1. If the authors have adequately addressed your comments raised in a previous round of review and you feel that this manuscript is now acceptable for publication, you may indicate that here to bypass the “Comments to the Author” section, enter your conflict of interest statement in the “Confidential to Editor” section, and submit your "Accept" recommendation.

Reviewer #1: All comments have been addressed

Reviewer #3: (No Response)

2. Is the manuscript technically sound, and do the data support the conclusions?

Reviewer #1: Yes

Reviewer #3: Yes

3. Has the statistical analysis been performed appropriately and rigorously? 

Reviewer #1: Yes

Reviewer #3: Yes

4. Have the authors made all data underlying the findings in their manuscript fully available?

Reviewer #1: No

Reviewer #3: Yes

5. Is the manuscript presented in an intelligible fashion and written in standard English?

Reviewer #1: Yes

Reviewer #3: Yes

6. Review Comments to the Author

Reviewer #1: Report on the first revision of “Automatic large-scale political bias detection of news outlets”

The paper presents a new way of analyzing news outlets with regards to a potential political bias in the presentation of news. The authors measure this bias as the deviation from a centered position along the typical left-right paradigm according to numerous parameters, which include not only socio-economic content, but also media type or geographical information on the outlets.

The paper has some obvious strengths, i.e the use of the data set, the algorithms and the relevance of the topic.

The main critical issues were that the concept of bias was not sufficiently clearly defined, the neglect to discuss the issue of multipolarity (the authors consider bipartisan as guiding principle sufficient), the unsatisfying motivation of the paper and the discussion of why the orientation along the chosen benchmarks is sufficient.

The revision has addressed all raised concerns, and the revision has made the overall topic not only more accessible, but also points to a wider applicability. I appreciate that the authors have also reworked the charts and provide data.

Given that, my comments are only minor:

- In lines 70ff you detail the different biases. It would give a good orientation to the reader if you could indicate directly here, which of these are taken into account in this study. In lines 214 you show that all four biases are more or less addressed. This more or less could be detailed, as the method is more reliable on size and tonality biases, isn’t it?

- You mention potential of selection bias due to covid (line 202); what risk do you see here with regards to your topic (relative and automated bias detection?). Is it possible that Covid eroded existing biases or aggravate them? Please elaborate, as my first reaction would be that Covid aggravated cleavages, it might also be argued that some biases eroded (as many countries took similar measures and many news outlets followed the general policy advice).

- Typos (e.g. rely on in line 131, simply in line 294)

- In line 346 you write that the main difficulty lies in the unexpected biases, I think this could be emphasized for media with tonality and bias changes (e.g. Twitter to X). then time can make a big difference. You used 2022 data, so this aspect does not play a role, but for time series analyses it might.

- I found it astonishing that factuality was a strong indicator according to the Shap-values. could you elaborate on this? Factuality should to my best understanding rather be neutral. This being not the case indicates that the model classifies either right or left as factual or not factual. Or does the model differentiate between different forms of factuality (this would be opening to arbitrariness).

Reviewer #3: Thank you to the authors for revising the manuscript and addressing the requests of all three reviewers.

I have had a read through the authors' response letter, the changes made to the revised manuscript, and the added Appendices and online supplement GiTHub repository, and I am confident that they meet the requirements for publication.

Therefore, I recommend to the Edtior the publication of this article in PLOS ONE.

7. PLOS authors have the option to publish the peer review history of their article (what does this mean?). If published, this will include your full peer review and any attached files.

Reviewer #1: No

Reviewer #3: No

---

## [Editor Report · Acceptance letter]

PONE-D-24-54904R1

PLOS ONE

Dear Dr. Rönnback,

I'm pleased to inform you that your manuscript has been deemed suitable for publication in PLOS ONE. Congratulations! Your manuscript is now being handed over to our production team.

Kind regards,

on behalf of

Dr. Shady Elbassuoni

Academic Editor

PLOS ONE